# APOBEC Mutagenesis in Cancer Development and Susceptibility

**DOI:** 10.3390/cancers16020374

**Published:** 2024-01-15

**Authors:** Alexandra Dananberg, Josefine Striepen, Jacob S. Rozowsky, Mia Petljak

**Affiliations:** 1Molecular Biology Program, Sloan Kettering Institute, Memorial Sloan Kettering Cancer Center, New York, NY 10065, USA; ald2042@med.cornell.edu (A.D.); jus4013@med.cornell.edu (J.S.); 2Medical Scientist Training Program, New York University Grossman School of Medicine, New York, NY 10016, USA; jacob.rozowsky@nyulangone.org; 3Department of Pathology, NYU Grossman School of Medicine, New York, NY 10016, USA; 4Perlmutter Cancer Center, NYU Grossman School of Medicine, New York, NY 10016, USA

**Keywords:** APOBEC deaminases, APOBEC mutagenesis, cancer, carcinogenesis, cancer susceptibility, cancer development, APOBEC mutational signatures

## Abstract

**Simple Summary:**

APOBEC cytosine deaminases represent potent mutational sources in over 50% of human cancers and are linked to tumor heterogeneity and therapy responses. However, the understanding of the contribution of APOBEC-mediated mutagenesis to cancer susceptibility and malignant transformation is still limited. The authors review the existing evidence for the impact of APOBEC mutagenesis on cancer development and identify gaps in related knowledge that need to be addressed.

**Abstract:**

APOBEC cytosine deaminases are prominent mutators in cancer, mediating mutations in over 50% of cancers. APOBEC mutagenesis has been linked to tumor heterogeneity, persistent cell evolution, and therapy responses. While emerging evidence supports the impact of APOBEC mutagenesis on cancer progression, the understanding of its contribution to cancer susceptibility and malignant transformation is limited. We examine the existing evidence for the role of APOBEC mutagenesis in carcinogenesis on the basis of the reported associations between germline polymorphisms in genes encoding APOBEC enzymes and cancer risk, insights into APOBEC activities from sequencing efforts of both malignant and non-malignant human tissues, and in vivo studies. We discuss key knowledge gaps and highlight possible ways to gain a deeper understanding of the contribution of APOBEC mutagenesis to cancer development.

## 1. Introduction

The AID (activation-induced cytidine deaminase)/APOBEC (apolipoprotein B mRNA editing enzyme catalytic subunit) family comprises eleven members as follows: AID, APOBEC1, APOBEC2, APOBEC3A, APOBEC3B, APOBEC3C, APOBEC3D/E, APOBEC3F, APOBEC3G, APOBEC3H, and APOBEC4. While APOBEC2 and APOBEC4 members lack known deaminase activities, other enzymes play pivotal roles in immune and metabolic processes [1,2,3,4,5,6,7]. Put briefly, AID-mediated cytosine deamination at immunoglobulin loci contributes to somatic hypermutation and antibody diversification; APOBEC1-mediated cytidine deamination generates a lower molecular weight form of apolipoprotein B (ApoB) in the small intestine, which is essential for triglyceride transport; and the APOBEC3 subfamily-mediated deamination of retroviral and viral cytosines and cytidines limits viral replication as part of an innate immune defense [1,2,3,4,5,6,7].

Certain AID/APOBEC enzymes emerged as prominent mutators in cancer. APOBEC1 and several APOBEC3 members (3A, 3B, 3C, 3D/E, 3F, 3H) preferentially deaminate cytosine bases in TC dinucleotides, which can lead to mutations in targeted cytosines [8,9,10,11,12,13]. Mutational signatures that are characterized by cytosine mutations in TC dinucleotides, which are reflective of APOBEC1 and relevant APOBEC3 activities in cancer genomes, have been detected in over 50% of cancers and most cancer types [14,15]. These include the single-base substitution (SBS) signatures of genome-wide non-clustered C > T (called signature “SBS2”) and C > G/A (SBS13) mutations in TC dinucleotides [14] as well as the signatures of clustered cytosine mutations in TC dinucleotides, *kataegis* (local strand-coordinated hypermutation) [15] and *omikli* (diffuse hypermutation) [16]. APOBEC3A and APOBEC3B are the only endogenous enzymes that are confirmed to induce these signatures in human cells [17,18], with indications that additional APOBEC deaminases may contribute to cancer mutagenesis [11,19,20]. Other mutational types linked to direct or indirect APOBEC activities include APOBEC3A-mediated small insertions, deletions [21] as well as substitutions in non-TC dinucleotide cytosines in certain palindromic sequences [22], APOBEC3G-mediated SBS mutations [20], a doublet base substitution signature [14], and structural as well as copy number variations [9,23,24,25]. Mutations associated with AID activities are also found in cancer genomes. AID-mediated cytosine deamination during somatic hypermutation directly induces mutations in C:G pairs in WRC (W = A or T base; R = A or G) motifs and indirectly contributes to mutations in T:A pairs [7]. Clustered mutations linked to direct and indirect AID activities (respectively, SBS84 and SBS85 in census COSMIC database signatures [14]) are frequently detected in immunoglobulin heavy chain variable region (IGHV) genes in chronic myeloid leukemia (CLL), multiple myeloma, and diffuse large B-cell lymphomas (DLBCLs) [26,27,28,29,30]. Although a non-clustered genome-wide signature (SBS9) was initially proposed to be associated with AID activity [31], recent data suggest otherwise [32]. AID activity primarily targets IGHV, but it can also affect other regions with a preferential targeting of ±2 kb from the transcription start sites of highly transcribed genes [26]. Mutations linked to off-target AID activities are generally higher in cases with a mutated IGHV [26,27,33]. Additionally, AID can also induce rearrangements that are frequently found in implicated cancer types [34].

AID has been implicated in cancer development and progression, with the related roles extensively reviewed recently [35]. However, the precise contributions of APOBEC deaminases to cancer evolution remains less well understood. While APOBEC enzymes may contribute to cancer evolution through non-mutagenic mechanisms [25,36,37], mutagenesis by these enzymes appears to have a more widespread impact on cancer [20,38,39,40,41,42,43,44]. APOBEC mutagenesis endures in vitro in human cancer cell lines [45], and its signatures often appear in the subclonal phylogenetic branches of primary tumors and metastatic cancers, with incidental observations of driver mutations in APOBEC-associated sequence contexts [46,47,48,49,50,51,52]. APOBEC3A and APOBEC3B have been linked to persistent cell evolution and therapy resistance in lung cancers [41,43,44], and APOBEC3B has been linked to resistance against androgen receptor (AR)-targeted therapy and Tamoxifen in prostate and estrogen receptor-positive (ER+) breast cancers [40,53]. Furthermore, in vivo studies suggest that APOBEC mutagenesis can promote tumor heterogeneity [20,54,55,56]. These data and others indicate that ongoing APOBEC mutagenesis likely plays a significant role in cancer progression, although further experimental validation is necessary, as we have discussed before [39]. However, the contribution of APOBEC mutagenesis to malignant transformation remains considerably less well understood. Here, we outline the existing evidence (Figure 1) for the role of APOBEC mutagenesis in carcinogenesis and cancer susceptibility, addressing key knowledge gaps and discussing possible ways forward in order to address them.

## 2. Germline Variants Implicating APOBEC Mutagenesis in Cancer Susceptibility

Several polymorphisms in genes encoding APOBEC enzymes have been associated with a differential risk of cancer. One such polymorphism, a 29.5 kb deletion of the consecutive 3′-end of the *APOBEC3A* and most of the *APOBEC3B* gene found on chromosome 22 (*A3AB* deletion), produces a hybrid sequence of *APOBEC3A* fused with the 3′-untranslated region (UTR) of *APOBEC3B* [57]. The prevalence of *A3AB* deletion varies across ethnicities (Southeast Asian, 36.9%; South American, 57.7%; African, 0.9%; European, 6%) [57] and has been associated with the increased risks of breast and ovarian cancers among Asian populations [58,59,60,61]. The links between the *A3AB* deletion and cancer risk in Europeans are conflicting [62,63,64,65], although carriers under 50 years of age show strong indications of an increased risk in lung and prostate cancers [64]. The mechanisms underpinning the links between the *A3AB* deletion and cancer risk are not well understood. Proposed explanations include the stabilized expression of the hybrid *APOBEC3A’s* transcript [66] and the increased nuclear localization of APOBEC3H conferred by the A3H-I haplotype found in relevant polymorphism carriers [11], both of which are predicted to generate more potent mutator enzymes. Indeed, breast cancers from carriers exhibit elevated mutational burdens of APOBEC-mediated SBS2 and SBS13 signatures, with more mutations in homozygous carriers compared with heterozygous ones [67,68]. Thus, the available data imply that polymorphism may confer an increased risk of cancer as a result of the overactivity of certain APOBEC enzyme(s) and a consequential increase in mutational burdens. 

Another polymorphism in the APOBEC-related locus, a single nucleotide polymorphism (SNP) rs1014971 (allele: T), has been associated with an increased risk of bladder cancer [69,70,71], *APOBEC3B* expression [69], and APOBEC-signature mutational burdens in bladder tumors [69]. This SNP is located in a putative long-distance enhancer region upstream from the *APOBEC3* cluster that can interact with the *APOBEC3B* promoter [72], potentially leading to an elevated *APOBEC3B* expression and APOBEC-associated mutational burdens [69]. Rs1014971 has also been associated with an increased cancer risk and the *APOBEC3B* expression in breast cancer [69]. However, breast cancers from polymorphism carriers do not display an increased number of APOBEC-mediated mutations [69]. It is possible that the increased expression of *APOBEC3B* associated with rs1014971 may contribute to breast cancer susceptibility through mutagenesis-independent mechanisms, some of which have been proposed’ as mechanisms with such a function before [36].

Overall, the link between germline polymorphisms in APOBEC loci and cancer risk strongly implicate APOBEC enzymes in cancer susceptibility. Experimental work is required to validate existing associations and to understand the underpinning mechanisms. Mechanistic insights combined with genome-wide association studies across larger and broader populations will be critical for understanding the differences in risks across populations and the cancer types conferred by polymorphisms in APOBEC-related loci.

## 3. Somatic Mutagenesis Implicating APOBEC Mutagenesis in Cancer Susceptibility

The hypothesis that APOBEC mutagenesis plays a role in carcinogenesis assumes that it induces driver mutations that contribute to malignant transformations. Somatic APOBEC-associated mutational signatures and driver mutations have been detected in many types of primary cancers [14,73]. However, it is often unclear whether the relevant driver mutations occur during or after a malignant transformation. Advances in DNA sequencing strategies [74] have enabled insights into APOBEC mutagenesis in non-malignant and pre-malignant human tissues, providing a glimpse into its activities before a malignant transformation is complete (summarized in Table 1). These emerging insights combined with data from cancer genome sequences are key to understanding the potential of APOBEC mutagenesis to contribute to associated cancer types.

The APOBEC-mediated SBS2 and SBS13 are either very rare or absent in hepatocellular carcinoma, testicular cancer, thyroid adenocarcinoma [14,94], and their respective non-malignant tissue types [76,77,89], implying that APOBEC mutagenesis may not play a significant role in the development of these cancer types. On the other hand, SBS2 and SBS13 are found in the majority of esophageal squamous cell carcinomas (ESCCs) but are rare in a normal esophageal epithelium, where they are detected in ~0–<5% samples [14,76,77,83,84,95,96]. A study on ESCC development found that these signatures are also rare (detected in ~4% of clones) and contribute low mutational burdens in cases of low-grade intraepithelial neoplasia (LGIN), with hypermutation only detected in cases of high-grade intraepithelial neoplasia (HGIN) where the relevant signatures presented themselves in ~25% of clones [83]. Nevertheless, both LGIN and HGIN clones harboring the APOBEC-mediated mutations exhibited TP53 biallelic loss and high levels of copy number alterations. These data suggest that APOBEC hypermutation occurs after acquiring initial TP53 mutations and is likely to not be a major contributor to genome instability in the early stages of ESCC [83,95]. Indeed, APOBEC-mediated mutational signatures are absent from the spectrum of TP53 mutations in ESCC [95]. Similarly, APOBEC-mediated signatures have been identified in pancreatic adenocarcinomas (~46%), endometrial adenocarcinomas (~11%), acute lymphoblastic leukemia (~11%), B-cell lymphomas (~10%), stomach adenocarcinomas (~19%), renal-cell carcinomas (~18%), liposarcomas (~95%), adrenocortical cancers (~71%) and biliary adenocarcinomas (48%) [14,97]; however, they have not been identified in cells from respective healthy tissue types [32,76,77,90,91]. It is possible that APOBEC mutagenesis becomes active only during the later stages of cancer development, as observed in the esophagus. In contrast, APOBEC-mediated mutations occur to varying degrees in the bronchial epithelium of a healthy lung [75,76] (detected in 11–78% of samples depending on the study), small [19,76,77] (14–73%) as well as colonic [80] (0.5%) intestinal crypts, bladder urothelium [78] (22%), and in cancers from corresponding tissue types, which are sometimes according to similar proportions [14,19,98]. APOBEC mutagenesis may thus play a more relevant role in the early stages of the development of these cancers. 

Overall, available data indicate that APOBEC mutagenesis can occur in non-malignant tissues, albeit with varying rates of prevalence across different tissue types, in a similar way to cancers [14]. Therefore, the potential contribution of APOBEC mutagenesis to carcinogenesis likely differs across tissue types due to the variable mechanisms and extents of APOBEC regulation and/or dysregulation. Future studies examining APOBEC-mediated mutational signatures and driver mutations across different stages of cancer cell evolution from larger cohorts are critical to determining the timing of such events and the extent to which APOBEC mutagenesis may contribute to the evolution of individual cancer types. Such studies should also encompass other tissues where APOBEC mutagenesis is prevalent in cancer, such as breast and ovary. Additionally, APOBEC mutagenesis is prevalent in cervical cancers as well as in head and neck squamous cell carcinomas (HNSCCs) where it has been proposed to contribute to cancer development as a consequence of misdirected activity against human papillomavirus (HPV), which is associated with the etiologies of these cancers [31,99]. Comparative analyses of samples affected and unaffected by the relevant viruses can provide further insight into the role of APOBEC deaminases in the etiology of HPV-associated cancer types. For example, APOBEC-mediated mutations are more common in HPV-positive HNSCCs where they have been linked to the generation of oncogenic *PIK3CA* mutations, unlike in HPV-negative HNSCCs [99]. When premalignant tissue samples are not routinely biopsied, such as those of HPV-positive HNSCC cancers, computational models that infer the phylogenetic relationship between tumor subclones can be utilized to predict the timing of the somatic events in carcinogenesis [100]. 

## 4. In Vivo Data Implicating APOBEC Mutagenesis in Carcinogenesis 

In vivo studies (summarized in Table 2) have offered valuable insight into the potential of APOBEC mutagenesis to contribute to cancer development. An early study found that transgenic mice and rabbits expressing rabbit *APOBEC1* in their livers developed hepatocellular carcinomas, while controls did not [101]. However, it is not clear whether APOBEC1 contributes to carcinogenesis through mutagenesis or an alternative mechanism. Importantly, the impact of APOBEC1 on human tissues is uncertain as it is typically expressed in the small intestine and duodenum but not the liver [19]. Additionally, APOBEC-mediated signatures have not been detected or do not contribute to many mutations in respective cancer types [14,94,98,102]. In vivo examination of the roles of APOBEC3 mutagenesis has largely relied on transgenic models due to the lack of most human *APOBEC3* ortholog genes in mice [103]. In models predisposed to colorectal cancer (Apc^Min^), the constitutive, ubiquitous expression of human tumor-like levels of *APOBEC3A* from the CAG promoter can promote the development of colorectal cancer [55]. Similarly, higher, likely transient, levels of *APOBEC3A* through an integrative transposable element can elevate liver cancer rates in Fah liver regeneration models, which are contingent on APOBEC3A catalytic activity [55,56]. In both models, induction of *APOBEC3A* inflicted APOBEC-associated mutations. Interestingly, expression of the other six *APOBEC3* paralogs, including *APOBEC3B*, failed to develop tumors in the Fah models [55]. However, a subsequent study found that the constitutive, ubiquitous expression of higher and human tumor-like *APOBEC3B* levels from CAG the promoter accelerates liver and lymphoma cancer formation as well as metastasis in non-predisposed animals [54]. Tumors from *APOBEC3B*-expressing mice accumulated a higher proportion of APOBEC-mediated SBS2 mutations, with all APOBEC3B-related phenotypes dependent on its catalytic activity, although the total mutational burdens in developed tumors were similar to those in controls. 

Collectively, these findings demonstrate that APOBEC3A- and APOBEC3B-mediated deamination can promote carcinogenesis. However, it remains less defined how well the relevant models reflect enzyme activities in human tissues. Inducing *APOBEC3A* and *APOBEC3B* for tumor levels may not accurately mimic the carcinogenesis arising in pre-malignant tissues given their generally lower expression in normal tissues [104,105,106]. Additionally, APOBEC mutagenesis can occur transiently in both human cancer cell lines [45] and non-malignant tissues [19,75] where it can, similarly to human cancers, only affect certain cellular lineages [19,50,75,78,79,80,107] and be infrequent over their lifetimes [19]. The constitutive, ubiquitous expression of APOBEC enzymes from a commonly used heterologous CAG promoter does not recapitulate these features since it separates expression from the regulatory mechanisms in human cells [54]. Indeed, depending on the level, duration, and model system, the expression of *APOBEC3B* can lead to various outcomes, including that of there being no overt tumor phenotypes [55,108], increased tumor rates [54], detriments to tumor development [43], or lethality [109]. This variability underscores the importance of inducing APOBEC enzymes under conditions that accurately replicate relevant human tissue settings.

**Table 2 cancers-16-00374-t002:** Existing APOBEC transgenic in vivo models. The “Mouse Model” column details the mouse strain upon which APOBEC induction was performed. The “APOBEC Induction Strategy” column outlines the specific APOBEC gene induced, localization, and timing of induction. The “Level of Induction” column indicates the level of APOBEC induction, while the ‘Phenotype’ column summarizes the resulting characteristics or effects upon induction.

Study	Mouse Model	APOBEC Induction Strategy (Gene, Localization, Induction)	Level of Induction	Phenotype
Yamanaka et al., 1995 [101]. PMID: 7667315	Wild-type (no cancer predisposition)	APOBEC1, ectopic (liver), stable	Overexpressed	*APOBEC1* expression causes the development of liver dysplasia and hepatocellular carcinomas. Transgenic animals contain transcripts that are extensively edited by APOBEC1.
Law et al., 2020 [55]. PMID: 32870257	Wild-type (no cancer predisposition)	APOBEC3A, ubiquitous, constitutive	Human tumor-like levels	*APOBEC3A* expression is insufficient for tumor initiation.
Boumelha et al., 2022 [108]. PMID: 35930804	Wild-type (no cancer predisposition)	APOBEC3B, ubiquitous, Cre-induced	Not reported	*APOBEC3B* expression does not induce tumors.
Boumelha et al., 2022 [108]. PMID: 35930804	*Kras^LSL^*^-G12D/+^, *Trp53*^fl/fl^-driven lung cancer model	APOBEC3B, ubiquitous, Cre-induced	Not reported	*APOBEC3B* expression does not increase tumor growth rate and fails to substantially increase clonal tumor mutational burden.
Boumelha et al., 2022 [108]. PMID: 35930804	Urethane-induced lung cancer model	APOBEC3B, ubiquitous, Cre-induced	Not reported	*APOBEC3B* expression does not increase tumor growth rate or the number of tumors per animal.
Law et al., 2020 [55]. PMID: 32870257	Adenomatous polyposis coli multiple intestinal neoplasia (Apc^min^)-driven colon cancer model	APOBEC3A, ubiquitous, constitutive	Human tumor-like levels	*APOBEC3A* expression in murine colon tissue increases tumorigenesis and APOBEC-associated mutations.
Law et al., 2020 [55]. PMID: 32870257	Adenomatous polyposis coli multiple intestinal neoplasia (Apc^min^)-driven colon cancer model	APOBEC3G, ubiquitous, constitutive	Not reported	*APOBEC3G* expression does not increase polyp formation.
Law et al., 2020 [55]. PMID: 32870257	Fumaryl-acetoacetate hydrolase (Fah) model for hepatocellular carcinoma (with sh*Tp53*)	APOBEC3(A-H), hydrodynamic transfer at 2 months	Not reported	*APOBEC3A* expression in murine liver tissue increases tumorigenesis and APOBEC mutations (SBS2 and SBS13). Other APOBEC3 paralogs fail to develop tumors.
Naumann et al., 2023 [56]. PMID: 37298259	Fumaryl-acetoacetate hydrolase (Fah) model for hepatocellular carcinoma	APOBEC3A, hydrodynamic transfer at 2 months	Not reported	APOBEC3A is capable of driving tumor development. Catalytic activity and DNA deamination (not RNA-editing) are required to promote tumor formation.
Durfee et al., 2023 [54]. PMID: 37797615	Wild-type (no cancer predisposition)	APOBEC3B, ubiquitous, constitutive	Human tumor-like levels	*APOBEC3B* expression accelerates rates of carcinogenesis, tumor heterogeneity, and metastasis in older animals. Transgenic animals display an increase in APOBEC-associated mutations, indels, and structural variations. APOBEC3B catalytic activity is required for all phenotypes.
Caswell et al., 2023 [43]. PMID: 38049664	*Tp53* WT, *EGFR*^L858R^-driven lung cancer model	APOBEC3B, ubiquitous, induced at tumor initiation	Not reported	*APOBEC3B* expression constrains tumorigenesis. Catalytic activity is required for the phenotype.
Liu et al., 2023 [20]. PMID: 36480186	*n*-butyl-N-(4-hydroxybutyl)nitrosamine (BBN)-induced bladder cancer model	APOBEC3G, ubiquitous, constitutive	Not reported	*APOBEC3G* expression promotes mutagenesis, genomic instability, and kataegis, leading to shorter survival in animals. A novel SBS signature is identified in animals expressing APOBEC3G.
Wormann et al., 2021 [25]. PMID: 35121902	Pdx1-Cre, *KRAS*^G12D^, *Tp53*^fl/fl^-driven pancreatic cancer model	APOBEC3A (truncated), ubiquitous, stable	Similar to A3A levels in human lymphocytes (physiological levels of A3A compared with human ones)	*APOBEC3A* expression leads to more aggressive tumors and metastasis independent of its canonical deaminase functions.
de la Vega et al., 2023 [109]. PMID: 38001542	Wild-type (no cancer predisposition)	APOBEC3B, ubiquitous, induced at 4 weeks s	In lungs, expression is within the range observed in human cancers. In the liver and pancreas, expression is comparable to human tumors with highest APOBEC3B levels and is associated with poor survival.	*APOBEC3B* expression leads to RNA editing and is lethal.

## 5. Conclusions

Overall, murine studies and insights from human genome sequencing efforts indicate that APOBEC enzymes can contribute to carcinogenesis, but further research is necessary to comprehend the mechanisms and extent of APOBEC mutagenetic involvement in the development of different human cancer types. Examination of APOBEC-associated mutational signatures and driver mutations across pre-malignant tissues and tissues reflective of the transitional stages during carcinogenesis, as well as development of models that closely resemble mutagenic APOBEC activities emerging from such studies, will be critical to assess their contribution to the development of different human cancer types.

## Figures and Tables

**Figure 1 cancers-16-00374-f001:**
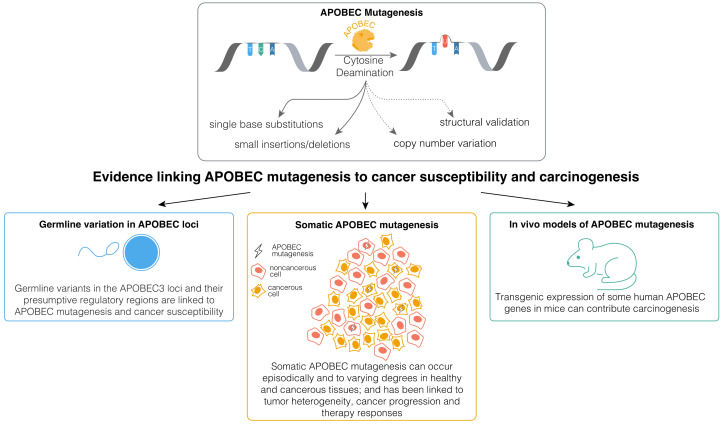
Existing lines of evidence implicating mutagenesis by APOBEC enzymes in cancer development and susceptibility.

**Table 1 cancers-16-00374-t001:** Prevalence of reported APOBEC-mediated mutational signatures (SBS2 and SBS13) in available genomes from non-malignant tissues. In studies that do not discuss the prevalence of SBS2 and SBS13 (designated with “*”), any number of mutations designated as SBS2 or SBS13 in a sample were considered to indicate their presence. In some instances, low burdens of SBS2 and SBS13 may thus represent false-positive calls.

Study	Tissue Type	Percent of Subjects with SBS2 and/or SBS13	Percent Samples with SBS2 and/or SBS13
Yoshida et al., 2020 [75]. PMID: 31996850	Lung: bronchus epithelium clones	16/16 subjects	493/632 (~78%)
Li et al., 2021 [76]. PMID: 34433965	Lung: bronchus epithelium microbiopsies	2/3 subjects	15/135 (~11%)
Moore et al., 2021 [77]. PMID: 34433962	Small intestine: epithelium crypts	2/3 subjects	36/49 (~73%)
Wang et al., 2023 [19]. PMID: 36702998	Small intestine: epithelium crypts	39/39 subjects	58/342 (~17%)
Li et al., 2021 [76]. PMID: 34433965	Small intestine: duodenum epithelium crypts	2/4 subjects	25/179 (~14%)
Lawson et al., 2020 [78]. PMID: 33004514	Bladder: urothelium microbiopsies	9/15 subjects	19/88 (~22%)
Olafsson et al., 2023 [79]. PMID: 37884686	Skin: epidermis microbiopsies	12/111 subjects (patients with Psoriasis) *	21/1182 (~2%) *
Lee-Six et al., 2019 [80]. PMID: 31645730	Colon: epithelium crypts	2/42 subjects	2/445 (~0.5%)
Olafsson et al., 2020 [81]. PMID: 32697969	Colon: epithelium crypts	4/46 subjects (ulcerative colitis, *n* = 28; Crohn’s disease, *n* = 18) *	26/446 (~6%) *
Lee et al., 2022 [82]. PMID: 35581206	Intestine: epithelium crypts	1/10 subjects (patients with Lynch syndrome)	10/107 (~10%)
Li et al., 2021 [76]. PMID: 34433965	Esophagus: epithelium microbiopsies	1/5 subjects	5/203 (~2%)
Chang et al., 2023 [83]. PMID: 38039962	Esophagus: epithelium microbiopsies	1/22 subjects	1/48 (~2%)
Chang et al., 2023 [83]. PMID: 38039962	Esophagus: low-grade intraepithelial neoplasia microbiopsies	1/9 subjects	1/23 (~4%)
Chang et al., 2023 [83]. PMID: 38039962	Esophagus: high-grade intraepithelial neoplasia microbiopsies	2/7 subjects	2/8 (~28%)
Martincorena et al., 2018 [84]. PMID: 30337457	Esophagus: epithelium microbiopsies	0/21 subjects	0/21 (0%)
Kakiuchi et al., 2020 [85]. PMID: 31853061	Colon: epithelium crypts	0/40 subjects (healthy, *n* = 22; ulcerative colitis, (*n* = 18)	0/101 (0%)
Robinson et al., 2022 [86]. PMID: 35803914	Intestine: epithelium crypts	0/10 subjects (patients with BER deficiency)	0/144 (0%)
Robinson et al., 2021 [87]. PMID: 34594041	Intestine: epithelium crypts	0/13 subjects (patients with POLE/POLD1 germline mutations)	0/109 (0%)
Brunner et al., 2019 [88]. PMID: 31645727	Liver: parenchyma microbiopsies	0/14 subjects (healthy, *n* = 5; alcohol-related liver disease, *n* = 4; non-alcoholic fatty liver disease, *n* = 5)	0/482 (0%)
Ng et al., 2021 [89]. PMID: 34646017	Liver: parenchyma microbiopsies	0/34 subjects (healthy, *n* = 5; alcohol-related liver disease, *n* = 10; non-alcoholic fatty liver disease, *n* = 19)	0/1590 (0%)
Osorio et al., 2018 [90]. PMID: 30485801	Bone marrow: clones (hematopoietic stem cells and multipotent progenitor cells)	0/5 subjects	0/18 (0%)
Machado et al., 2022 [32]. PMID: 35948631	Blood: clones (native B, memory B, CD4+ and CD8+ native T cells, CD4+ and CD8+ memory T cells)	0/7 subjects	0/717 (0%)
Moore et al., 2020 [91]. PMID: 32350471	Endometrium: gland microbiopsies	28 subjects	0/292 (0%)
Coorens et al., 2021 [92]. PMID: 33692543	Placenta: bulk tissue	0/37 subjects	0/86 (0%)
Buhigas et al., 2022 [93]. PMID: 36131292	Prostate: bulk tissue	0/37 subjects	0/51 (0%)
Li et al., 2021 [76]. PMID: 34433965	Pan-tissue microbiopsies	Colon: 0/5 Gastric cardia: 0/3 Liver: 0/5 Pancreas: 0/5 Rectum: 0/4 Stomach: 0/3	Colon: 0/246 (0%) Gastric cardia: 0/126 (0%) Liver: 0/248 (0%) Pancreas: 0/249 (0%) Rectum: 0/188 (0%) Stomach: 0/188 (0%)
Moore et al., 2021 [77]. PMID: 34433962	Pan-tissue microbiopsies	Adrenal gland: 0/1 Appendix: 0/1 Bladder: 0/1 Bronchus: 0/1 Colon: 0/7 Heart: 0/1 Kidney: 0/2 Liver: 0/2 Esophagus: 0/2 Pancreas: 0/2 Prostate: 0/2 Skin: 0/2 Small bowel: 0/3 Stomach: 0/2 Testis: 0/13 Thyroid: 0/1 Ureter: 0/1 Visceral fat: 0/1	Adrenal gland: 0/15 (0%) Appendix: 0/20 (0%) Bladder: 0/7 (0%) Bronchus: 0/22 (0%) Colon: 0/50 (0%) Heart: 0/6 (0%) Kidney: 0/19 (0%) Liver: 0/27 (0%) Esophagus: 0/30 (0%) Pancreas: 0/19 (0%) Prostate: 0/20 (0%) Skin: 0/14 (0%) Small bowel: 0/49 (0%) Stomach: 0/2 (0%) Testis: 0/209 (0%) Thyroid: 0/31 (0%) Ureter: 0/4 (0%) Visceral fat: 0/5 (0%)

## Data Availability

No data was generated in the Commentary. The data summarized in Table 1 and Table 2 was sourced from the cited papers.

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
