# Peer review of "APOBEC Mutagenesis in Cancer Development and Susceptibility"

_cancers, 2024, doi:10.3390/cancers16020374_

Round 1

Reviewer 1 Report

Comments and Suggestions for Authors

The manuscript by Dananberg et al is devoted to the quickly unfolding field of APOBEC-induced carcinogenesis. Overall, this is a good summary of the (especially, most recent) findings and studies and deserves to be published, will be of use to the community.

I have one rather major comment. It is said in the introduction that APOBEC family consists of 10 members. This is not true. The enzyme superfamily is called AID/APOBEC. It has certain sub-families (i.e. APOBEC1, AID, APOBEC2, etc). In humans, there are indeed 10 genes called "APOBEC", plus an AID, which is, once again, bioinformatically and functionally belongs to the same family of enzymes. Historically, AID is not called APOBEC, but that's a terminology issue, not relevant to biology.

So, my suggestion is, to re-write the introduction so that the readers won't be misled. If authors don't want to discuss AID (which is clearly associated with many cancers), just point it out. But excluding AID from the discussion altogether, as well as failure to note that all APOBEC3's belong to a single subfamily (simply put, they are more similar to each other than to other APOBECs) can propagate a wrong understanding of AID/APOBEC superfamily, especially in readers who are new to the field. 

Author Response

We thank the Reviewer for taking the time to read our manuscript and providing valuable feedback. We agree that AID deaminase plays a crucial role in cancer. However, due to the word limit, we were unable to provide a comprehensive overview of both APOBEC subfamilies and AID's role in cancer susceptibility and carcinogenesis. Additionally, while such roles of AID have been recently reviewed elsewhere (Jiao et al, Front Immunol., 2023), a comprehensive summary of the role of APOBEC subfamilies in cancer initiation has not been provided to the best of our knowledge. Furthermore, there are ongoing debates and questions regarding some of the associations and models used to link the APOBEC family to various aspects of carcinogenesis, as mentioned in the text. Therefore, we decided to focus our review on APOBEC subfamilies to highlight the current state of the field and encourage further research.

We acknowledge that our original descriptions may have caused confusion and appreciate the Reviewer's feedback. As a result, we have updated the introduction (changes are highlighted below) to provide a clearer description of the AID/APOBEC family:

The AID (activation-induced cytidine deaminase) / APOBEC (apolipoprotein B mRNA editing enzyme catalytic subunit) family comprises eleven members: AID, APOBEC1, APOBEC2, APOBEC3A, APOBEC3B, APOBEC3C, APOBEC3D/E, APOBEC3F, APOBEC3G, APOBEC3H, and APOBEC4. While APOBEC2 and APOBEC4 members lack known deaminase activities, other enzymes play pivotal roles in immune and metabolic processes1–7. Briefly, AID-mediated cytosine deamination at immunoglobulin loci contributes to somatic hypermutation and antibody diversification; APOBEC1-mediated cytidine deamination generates a lower molecular weight form of apolipoprotein B (ApoB) in the small intestine essential for triglyceride transport; and APOBEC3 subfamily-mediated deamination of retroviral and viral cytosines and cytidines limits viral replication as part of innate immune defense1–7.

Certain AID/APOBEC enzymes emerged as prominent mutators in cancer. APOBEC1 and several APOBEC3 members (3A, 3B, 3C, 3D/E, 3F, 3H) preferentially deaminate cytosine bases at TC dinucleotides, which can lead to mutations at targeted cytosines8–13. Mutational signatures characterized by cytosine mutations at TC dinucleotides reflective of APOBEC1 and relevant APOBEC3 activities in cancer genomes have been detected in over 50% of cancers and most cancer types14,15. These include single-base substitution (SBS) signatures of genome-wide non-clustered C>T (signature ‘SBS2’) and C>G/A (‘SBS13’) mutations at TC dinucleotides14, as well as signatures of clustered cytosine mutations in TC dinucleotides, kataegis (local strand-coordinated hypermutation)15 and omikli (diffuse hypermutation)16. APOBEC3A and APOBEC3B are the only endogenous enzymes confirmed to be inducing these signatures in human cells17,18, with indications that additional APOBEC deaminases may contribute to cancer mutagenesis11,19,20. Other mutational types linked to direct or indirect APOBEC activities include APOBEC3A-mediated small insertions, deletions21, and substitutions at non-TC dinucleotide cytosines in certain palindromic sequences22, APOBEC3G-mediated SBS mutations20, a doublet-base substitution signature14, as well as structural and copy number variations9,23–25. Mutations associated with AID activities are also found in cancer genomes. AID-mediated cytosine deamination during somatic hypermutation directly induces mutations at C:G pairs in WRC (W = A or T base; R = A or G) motifs and indirectly contributes mutations at T:A pairs7.  Clustered mutations linked to direct and indirect AID activities (respectively, SBS84 and SBS85 in census COSMIC database signatures14) are frequently detected in immunoglobulin heavy-chain variable region  (IGHV) genes in chronic myeloid leukemia (CLL), multiple myeloma, and diffuse large B-cell lymphoma (DLBCL)26–30. Although a non-clustered genome-wide signature (SBS9) was initially proposed to be associated with AID activity31, recent data suggest otherwise32. AID activity primarily targets IGHV, but it can also affect other regions with a preferential targeting ±2 kb from the transcription start sites of highly transcribed genes26. Mutations linked to off-target AID activities are generally higher in cases with mutated IGHV33–35. Additionally, AID can also induce rearrangements frequently found in implicated cancer types36.

AID has been implicated in cancer development and progression, with its roles extensively reviewed elsewhere
37. However, the precise contributions of APOBEC deaminases in cancer evolution remains less well understood. While APOBEC enzymes may contribute to cancer evolution via non-mutagenic mechanisms25,38–42, mutagenesis by these enzymes appears to have a more widespread impact on cancer20,38,43–48. APOBEC mutagenesis endures in vitro in human cancer cell lines49, and its signatures often appear in subclonal phylogenetic branches of primary tumors and metastatic cancers, with incidental observations of driver mutations in APOBEC-associated sequence contexts50–56. APOBEC3A and APOBEC3B have been linked persistent cell evolution and therapy resistance in lung cancers45,47,48, and APOBEC3B to resistance to androgen receptor (AR)-targeted therapy and Tamoxifen in prostate and estrogen receptor-positive (ER+) breast cancers44,57. Furthermore, in vivo studies suggest that APOBEC mutagenesis can promote tumor heterogeneity20,58–60. These and other data indicate that ongoing APOBEC mutagenesis likely plays a significant role in cancer progression, albeit further experimental validation is necessary, as we discussed before43. However, the contribution of APOBEC mutagenesis to malignant transformation remains considerably less well understood. Here, we outline the existing evidence for the role of APOBEC mutagenesis in carcinogenesis and cancer susceptibility, address key knowledge gaps, and discuss possible ways forward to address them.

Reviewer 2 Report

Comments and Suggestions for Authors

APOBEC mutagenesis are associated with tumor heterogeneity, ongoing tumor evolution, and response to treatment. However, the role of APOBEC in the development of cancer is not fully understood, and certain APOBEC genotypes, such as APOBEC3A/3B deletions, have been linked to an increased risk of multiple cancers. This suggests that APOBEC-mediated mutations may contribute to a higher risk of cancer. Furthermore, APOBEC-mediated mutations have been found in precancerous tissues, but their occurrence varies depending on the tissue type. This indicates that APOBEC-mediated mutations may have different effects on the development of various cancer types. Several APOBEC transgenic mouse models have shown that APOBEC-mediated mutations can promote the development of cancer cells. However, it is important to note that these models may not fully replicate the situation in the human body. While existing data support the idea that APOBEC-mediated mutations can promote carcinogenesis, further research is needed to understand the mechanisms involved and the extent of their effects on different cancer types, particularly in relation to precancerous tissues in humans. This review provides a comprehensive summary of the current understanding of the carcinogenic effects of APOBEC. It also suggests areas for further research and is an important reference for researchers in the field. In terms of modifications

1,The introductory section could be slightly expanded to provide some background information on the mutational characteristics of APOBEC. Add some reference papers, e.g., " APOBEC Alteration Contributes to Tumor Growth and Immune Escape in Pan-Cancer".

2, In addition, the quality of the figures and tables could be further improved to enhance the visualisation of the article.

Author Response

  1. We thank the Reviewer for their positive feedback and the suggestions. While the word limit constrained us from expanding the introductory section, we would like to emphasize that our original manuscript provides a thorough overview of the characteristics of mutational patterns induced by APOBEC enzymes (in introduction), as well as prevalence of somatic mutagenesis by APOBEC enzymes in cancer and normal tissues (section ‘Somatic mutagenesis implicating APOBEC mutagenesis in cancer susceptibility’). Additionally, we appreciate the Reviewer's suggestion to include the relevant reference on the expression of APOBEC genes in normal and cancer samples, as well as correlations between APOBEC expression levels and prognostic values in cancer types. While our review specifically focuses on the evidence implicating APOBEC enzymes in cancer development and carcinogenesis, we have added the reference at the fitting place in the text:

    Inducing APOBEC3A and APOBEC3B to tumor levels may not accurately mimic carcinogenesis arising in pre-malignant tissues, given their generally lower expression in normal tissues(Roberts et al. 2013; Burns et al. 2013; Guo et al. 2022).

  2. We appreciate the Reviewer’s comments that helped us enhance our manuscript. We have now improved the table styles for clarity and enhanced visualization, as well as added a figure summarizing high-level evidence implicating APOBEC enzymes in cancer susceptibility and carcinogenesis, which we describe in text (Figure 1).